# Motivators of the Intention of Wine Tourism in Baja California, Mexico

**DOI:** 10.3390/foods13223651

**Published:** 2024-11-16

**Authors:** Sandra Nelly Leyva-Hernández, Arcelia Toledo-López

**Affiliations:** 1Facultad de Ingeniería y Negocios San Quintín, Universidad Autónoma de Baja California, San Quintín 22930, Mexico; sandra.n.leyva@gmail.com; 2Centro Interdisciplinario de Investigación para el Desarrollo Integral Regional Unidad Oaxaca, Instituto Politécnico Nacional, Oaxaca 71230, Mexico

**Keywords:** traditional gastronomy, intention of wine tourism, PLS-SEM, Baja California, Mexico

## Abstract

The present study aimed to analyze which variables have the most significant effect and importance in analyzing wine tourism intention in Baja California, Mexico, using the stimulus–organism–response model. An exploratory and cross-sectional study collected a sample of 728 adult wine consumers from Baja California. Data analysis was carried out using structural equation modeling by partial least squares and analysis of the importance–-performance map. Traditional gastronomy was the main predictor of wine tourism intention in Baja California, Mexico, and according to the importance–performance map, it is the most essential variable in the analysis. In addition, the study results showed that identity mediates the relationship between electronic word of mouth and wine tourism intention and between traditional gastronomy and wine tourism intention. The study employs an importance–performance mapping analysis that has yet to be used in wine tourism analysis and proposes a stimulus (electronic word of mouth, traditional gastronomy)–organism (identity)–response (wine tourism intention) model to broaden understanding of the phenomenon. Its findings and methodology can serve as a valuable template for future research, offering a blueprint that can be replicated in regions like Baja California. This research has significant consequences for creating marketing plans in the wine tourism sector.

## 1. Introduction

In Latin America in 2023, the countries with the highest wine production are Chile, Argentina, and Brazil, where Chile ranks first with a production of 11,000 hectoliters [1]. Mexico’s wine industry, while not as large as that of other countries, is a significant contributor to the country’s economy. It is the second largest employer in the primary sector, providing over 500,000 jobs. Additionally, the industry produces an impressive four million cases of wine annually [2,3].

Baja California has 70% of Mexico’s wine production, with the remainder produced in Zacatecas, Coahuila, Querétaro, Aguascalientes, Sonora, and Chihuahua [4]. However, the sector faces problems regarding its growth and competitiveness as it only satisfies 30% of domestic consumption, in addition to the fact that sales volume has fallen; in 2020, it had lower sales than in 2016, and for the consumption of alcoholic beverages in Mexico it is ranked fourth. All this is due to low profitability, scarce research and technology transfer, and lack of advice on the transformation, integration, and strengthening of activities [5]. Wine tourism has the potential to significantly enhance Baja California’s competitiveness by showcasing the region’s top-quality wines and fostering its overall development [6]. This underscores the importance of comprehending the critical drivers of wine tourism from visitors’ perspectives. Such insights will be instrumental in crafting innovative tourism development initiatives tailored to the specific needs and preferences of tourists in the area.

The wine industry has seen a rise in wine tourism, with entrepreneurs expanding their business models to offer tourism services alongside wine sales [7]. Wine tourism encompasses visiting wineries, wine exhibitions, wine tasting, and enjoying the wine landscape [8]. Wine tourism can be considered a development path for rural wine-producing regions [9,10], as in the case of Baja California.

Authors such as Ramos [11] highlight that to enhance the understanding of wine tourism in Baja California, Mexico, it is essential to explore the impact of electronic word of mouth (E-WOM). Furthermore, in Mexico, wine plays a central role in the country’s culture and identity as it has been part of its history and gastronomy through its flavors, customs, and traditions [12,13]. This also highlights the role of identity and traditional gastronomy in the analysis of wine tourism in Baja California.

The stimulus–organism–response model, a key tool in understanding behaviors, elucidates how external cognitive and affective influences, termed as stimuli, impact the internal state of an individual, leading to a specific behavior [14]. This model posits that external factors predict the internal state, which in turn predicts behavior. Therefore, using the stimulus–organism–response model, this research aims to determine the variables with the most significant effect and importance in the intention of wine tourism in Baja California. In this way, the study contributes to the understanding of wine tourism in rural regions such as Baja California, Mexico.

## 2. Literature Revision

The stimulus–organism–response model posits that external factors predict the internal state, which in turn predicts behavior [14]. Analyzing the food consumption behavior model explains the motivators that lead to behavior [15,16]. Lee et al. [15] describe how intrinsic and extrinsic characteristics (stimuli) modify the attitude of the consumer (organism) and consequently explain the purchasing behavior of organic foods. Also, Leyva-Hernández et al. [16] explain how external stimuli modify the identity of the consumer (organism), and in turn this influences socially responsible consumption (response). However, it also proves that identity acts as a mediating variable between the relationship of stimuli and response. Another example where this model is considered as a framework for a study is that of Manthiou et al. [17], who analyze how the physical environment (stimulus) influences the cognitive and emotional perspectives of the consumer (organism) and consequently leads to behavior toward the environment (response). Using this framework, it is possible to propose an analysis of the intention of wine tourism based on the external stimuli and the internal state of the wine tourist.

Traditional gastronomy can be considered an external stimulus that leads to wine tourism because the wine tourism experience encompasses both gastronomic and cultural dimensions [18]. Vibrant celebrations take place in wine-producing regions that highlight local food and wine, elevating the experience with a rich tapestry of flavors and traditions [19]. In Mexico, wines pair recipes with traditional ingredients, highlighting the flavors of both and reflecting culinary diversity [13]. Each wine region has a dedicated event for wine appreciation that integrates gastronomic elements.

In turn, E-WOM can be considered another external stimulus because E-WOM equips tourists with valuable information, empowering them to choose their wine tourism destination [8]. When making purchasing decisions, wine tourists are swayed by online reviews and recommendations, as sifting through and comparing this information helps alleviate any uncertainty about their chosen destination [9]. Consequently, digital platforms are crucial in ensuring and enhancing the wine tourism experience [18]. Effective communication is crucial in shaping the wine tourism experience, considering that wine is a comprehensive product [9]. Consequently, word of mouth is a potent tool for driving tourism engagement in the wine industry. It enables consumers to assess a company’s appeal and reputation before embarking on their trip [7,9].

In the analysis of wine tourism, identity emerges as a central factor, which was considered in this research as the internal state of the wine tourist. It is closely tied to individuals’ unique self-image and their perceptions about something [20]. The development of identity is shaped by a complicated interaction between socialization and the distinct culture of the region. This culture includes a vibrant array of local customs, traditions, and values that characterize the community. These cultural aspects nurture a feeling of belonging and establish a collective story that links individuals to their environment, making the experience of identity both personal and communal [21].

Wine tourism has successfully crafted a unique image of local production and gastronomy [22]. Moreover, the internal identity of wine tourists is intricately linked to the culture and landscape of wine, which in turn shapes the image of the destination [23]. This is how identity, with its profound influence, guides decision-making about which wine tourism destination to visit, creating a personal connection to the choices made. In this sense, the research proposes a model where the stimuli are E-WOM and traditional gastronomy as variables external to the wine tourist, and identity is considered the internal state (organism) that leads to a response such as the intention of wine tourism, and the mediating analysis of identity is tested.

## 3. Materials and Methods

An exploratory study was conducted using a cross-sectional approach, following the guidelines of the Declaration of Helsinki. Seven hundred and twenty-eight data records were collected from visitors to wineries in the state of Baja California, Mexico. Data analysis was performed using partial least squares structural equation modeling (PLS-SEM) and importance–performance map analysis (IPMA) at the variable and indicator levels. PLS-SEM is advantageous because it has high statistical power despite a small sample size and does not suffer from representativeness issues; however, larger sample sizes enhance precision [24], as in this study.

Data were collected using a structured questionnaire through convenience sampling until we had obtained the minimum required to have a statistical power of 80%. The selection criteria for participants were that they were 18 years of age or older, resided in Baja California (see Figure 1), and had visited a winery in Baja California, Mexico. Data were collected using a Google Forms form from those participants who gave their consent to participate in the study. The sample was predominantly female (60.85%), which could represent a gender bias in the research. However, when there are two groups, it is enough for each group to have at least 30 cases to be representative if comparisons need to be made between groups, for example with more robust analyses such as structural equation modeling [25,26,27]. No comparisons were made between gender groups in the research, and in any case the male group, with the lowest number of cases (285), exceeded 30 data subjects. Therefore, each group was considered representative in the research. The respondents were distributed across Generation Z (65.1%), Generation Y (23.5%), Generation X (8.1%), and Baby Boomers (3.3%). In terms of education, the sample was diverse, with 2.5% having a primary degree, 12.9% having a secondary degree, 45.6% having a diploma degree, 29.4% having a bachelor’s degree, and 9.6% having postgraduate studies.

Cohen’s statistical power tables were used to calculate the minimum sample size, which, according to Benitez et al. [29], can be determined under this criterion when the data analysis is carried out by PLS-SEM. Thus, with three predictors, a small effect size for a conservative approximation, and a significance level of 0.05, the minimum size required was 550 [29,30].

Four latent variables (E-WOM, traditional gastronomy, identity, and wine tourism intention) were analyzed, which were part of the research model. These variables were measured using a 7-point Likert scale. The E-WOM variable was adapted from the study by Zayed et al. [31]. The traditional gastronomy variable was adapted from the research by Hernández-Rojas et al. [32]. The identity variable was adapted from the study by Leyva-Hernández et al. [16], and the wine tourism intention variable was adapted from the research by Leyva-Hernández et al. [33]. The measurements of each variable are presented in Table 1.

### Data Analysis

The data analysis by PLS-SEM consisted of three stages: the evaluation of the measurement model and the structural model as proposed by Hair et al. [34], and the analysis of the importance–performance map. Prior to the evaluation by PLS-SEM, a thorough verification process was undertaken to ensure the validity of the data, including the confirmation that the asymmetry and kurtosis values were in a range from −1 to +1, as recommended by Hair et al. [24]. The estimation of the model was carried out using the PLS algorithm and bootstrapping consistent with 5000 subsamples, by PLS Predict and by IPMA using version 4 of the SmartPLS software [35].

In the first stage, the external loadings of the variables’ indicators, the construct’s reliability, and the convergent and discriminant validity were verified [29]. In the second stage of the analysis, the model’s collinearity was verified, and the path coefficients and their significance, the coefficients of determination (R^2^) and prediction (Q^2^), and the effect sizes (f^2^) were determined [34]. In the third stage, the steps indicated by Hair Jr et al. [26] were followed. The requirements for the analysis of the importance–performance map were verified: that the indicators use an equidistant scale, which in the case of the research are 7-point Likert-type scales with a neutral point, that the indicator coding has the same orientation, and that the weights of the indicators are positive. In addition, the performance values were calculated, followed by the importance values, and thus the importance–performance map was created at the variable level. This map was also extended to the indicator level.

## 4. Results

The first stage involved a meticulous evaluation of the measurement model. Loads higher than 0.708 are retained to ensure the model’s robustness [29]. As a result, item GTR3 of the latent variable traditional gastronomy was eliminated. Table 2 presents the external loadings of the items validated for this study.

In evaluating the variables’ reliability, it was verified that they had values greater than 0.8 to have strict reliability but less than 0.95 to avoid redundancy [29]. For evaluating convergent validity, values equal to or greater than 0.5 are considered acceptable [34], and, in the case of the variables of this study, they met this criterion (Table 3).

The discriminant validity of the latent variables was verified using the Heterotrait–Monotrait Ratio (HTMT) values of the relationships of the variables. Values between two constructs less than 0.85 were considered [34]. The HTMT values are shown in Table 4. As well, the value (0.03) of the index of the Standardized Root Mean Square Residual (SRMR) was under 0.08, indicating a good fit of the model [29].

In the second stage, it was verified that all the VIF values of the structural model were less than 3 to prove no multicollinearity between the latent variable constructs [34]. In addition, the path coefficients (β) were calculated, and their significance is shown in Table 5 and Figure 2. The R^2^, f^2^, and Q^2^ values were calculated [29,34].

All relationships were significant. E-WOM positively and significantly influenced wine tourism intention (β = 0.199, *p* = 0.000). Traditional gastronomy positively and significantly influenced wine tourism intention (β = 0.471, *p* = 0.000). The effect of this stimulus is more significant than the previous one. E-WOM had a positive and significant impact on identity (β = 0.454, *p* = 0.000) and traditional gastronomy (β = 0.228, *p* = 0.000). And identity positively and significantly influenced wine tourism intention (β = 0.282, *p* = 0.000).

In addition, the steps proposed by Hair et al. [24] were followed to assess mediation. In the first step, the significance of the indirect effect was tested, a significant effect indicating mediation. It was proven that identity acted as a mediating variable between the stimuli (E-WOM and traditional gastronomy) and the response (wine tourism intention) since its indirect effect in the cases was significant, as pointed out by Nitzl et al. [36]. Therefore, identity significantly mediated the relationship between E-WOM and wine tourism intention (β = 0.128, *p* = 0.000) and the relationship between traditional gastronomy and wine tourism intention (β = 0.064, *p* = 0.000). The second step was to test the significance of the direct effect. In both cases, the direct effect was significant, which shows partial mediation, according to Carrión et al. [37].

The effect sizes in Table 5 show that the only large effect size is the relationship between traditional gastronomy and wine tourism intention, being more significant than 0.35 [29]. Meanwhile, the R^2^ values for wine tourism intention were moderate (R^2^ = 0.512), and for identity they were small (R^2^ = 0.512), according to the criteria of Benitez et al. [29]. The Q^2^ value was 0.313 for identity and 0.502 for wine tourism intention. This indicates that the independent variables of the model have a predictive power of almost 50% in the wine tourism intention.

As a result of the third stage, the importance–performance maps were obtained at the variable and indicator levels, as shown in Figure 3 and Figure 4. These maps allow us to know how the performance of dependent variables can be improved, in this case the intention to engage in wine tourism, by analyzing four areas. The x-axis is constructed with the importance values and the y-axis with the performance values. According to Hair Jr [26], the area located in (1) the lower right section, which corresponds to high importance and low performance, is the one with the greatest opportunity to improve the dependent variable, followed by the area of (2) the upper right section, which corresponds to high importance and high performance, then by the area of (3) the lower left section, which corresponds to low importance and low performance, and finally the area located in (4) the upper left, which corresponds to low importance and high performance.

In Figure 3, there are no variables in the greatest opportunity category, but the traditional gastronomy variable is in the upper right section, which is the one with the best performance and greatest importance. In addition, the E-WOM and identity variables are in the section with priority 3 for improvement since they are in the lower left section.

At the indicator level, the indicators with the most significant importance and performance are also those that measure traditional gastronomy (Figure 4). These are in the section with priority 2 for improvement and are GTR1, GTR2, and GTR4. In the section with priority 3, which is below average performance and importance, are the other indicators. Notably, EWO2 “I check online reviews of food and wine places to help me choose my meal” is above the average performance and therefore is in the section with priority 4 for improvement, indicating its null potential for further enhancement.

## 5. Discussion

The results show that the stimulus–organism–response model is a framework that explains the intention of wine tourism since its explanatory and predictive power is greater than 50%. It is proven that the stimuli (E-WOM and traditional gastronomy) affect the organism (identity), which in turn affects the response (wine tourism).

The variable that had the most significant effect on the intention to visit a winery was traditional gastronomy in the tested model. The recognition of Baja California as a gastronomic reference is the one that most influences the decision of wine tourists, followed by how they perceive themselves as wine connoisseurs and, finally, online wine reviews.

Also, it was found that when identity intervenes, it strengthens the effect of traditional gastronomy on the intention to visit a winery and the effect of E-WOM on the intention to visit a winery. Identity acts as a mediator of relationships. When wine tourists recognize the destination as a gastronomic reference and consider themselves wine connoisseurs, they are more willing to visit a winery. In turn, wine tourists who have read a wine review and consider themselves wine connoisseurs will be more likely to visit the winery.

Furthermore, the variable with the most significant potential for improving wine tourism in the tested model was traditional gastronomy, according to the importance–performance maps, which has implications for winery management. It is suggested that winery managers prioritize improving the image of traditional gastronomy to increase tourism in their wineries. They can hold events that promote the integration of wine pairing with local cuisine. Producers can also work with other entrepreneurs and public policy decision-makers to carry out a city marketing program in Baja California to highlight Baja California’s gastronomic heritage and wine production. Thus, since traditional gastronomy is the variable that can most improve wine tourism and positively affect it, producers’ efforts should prioritize improving this.

Despite not being native to Mexico, wine production has been enriched with local flavors, traditions, and ingredients, making it an integral part of the country’s culture and history [12,13,19]. This unique cultural context explains the significance of pairing wines with local dishes and the recognition of Baja California as a region with a rich gastronomic heritage. This cultural richness is a key factor in the importance of traditional gastronomy as the primary element influencing a consumer’s decision to visit a Baja California winery.

However, identity and E-WOM also positively and significantly affected the intention to visit. Therefore, attention should be paid to these variables as complementary to the strategies that encourage regional wine tourism. On the one hand, the plan for traditional gastronomy, a cornerstone of wine tourism, should be combined with the inclusion of tasting explanation services for wine tourists to increase their intention to visit the winery. And on the other hand, the inclusion of these tasting services should also be accompanied by digital infrastructure of the wine company, such as social networks and websites, where customers can make recommendations and comments about the experience they had during their visit so that future visitors can review the information.

The effect size of identity on wine tourism intention was small. However, this should not discourage winery managers and owners. For wine tourists, establishing a sense of place identity is a multifaceted journey that involves recognizing and engaging with distinct symbols, captivating experiences, and rich sensory attributes that define a particular destination [38]. These elements are not yet fully developed in Baja California, but there is a clear path forward. Winery managers and owners can generate programs that not only integrate the visit’s cultural and symbolic experience but also place a strong emphasis on sensory engagement. This will increase consumers’ identity and thus promote wine tourism.

The effect size of E-WOM, like that of identity, was small, suggesting that although it is not a factor that has a high relevance, it does affect wine tourism. As Ramos et al. [11] point out, the effect of prior knowledge, such as word of mouth (WOM) and electronic word of mouth (E-WOM), is an area that requires further research and presents an exciting opportunity for scholarly exploration. The experience prior to the visit is crucial for the intention to revisit and the degree of enjoyment during the visit. Therefore, as previously mentioned, wineries must have a digital infrastructure so that their visitors can rate their experience.

Our results also demonstrate the practical implications of the importance–performance map in understanding the variables that are most important in the analysis of wine tourism. While these maps have traditionally been used in the hospitality and food service sector to gauge service quality, customer satisfaction, and loyalty [39,40,41], our research shows that they are also invaluable in confirming the importance of traditional gastronomy in the analysis of wine tourism intention. This finding underscores the relevance and applicability of our research in the fields of tourism, hospitality, and gastronomy.

## 6. Conclusions

This research, uniquely, identified the variables with the most significant effect and importance in analyzing the intention to visit a winery from the stimulus–organism–response model by means of PLS-SEM and an importance–performance map. Using a little-studied theoretical framework, by exploring how external stimuli positively affect the consumer’s response, and, in turn, how their internal state strengthens this effect when analyzing wine tourism, this research significantly advances the understanding of existing studies.

The study analyzed how external stimuli affect the internal state of the wine tourist, and this, in turn, leads to an intention to visit. It was proven that the two stimuli, E-WOM and traditional gastronomy, have a positive and significant effect on the internal state of the identity of the wine tourist, and in turn, the identity of the wine tourist has a positive and significant impact on the intention to visit a winery, in addition to its participating as a mediating variable of the stimuli and the response. However, the effects of the stimuli on the wine tourism intention were also tested, and it was found that traditional gastronomy has the most significant impact on the wine tourism intention.

These results, consistent with those obtained by means of an importance–performance map, have significant practical implications. Traditional gastronomy was identified as the variable with the most significant importance in the analysis of wine tourism, suggesting that efforts in this area could significantly improve wine tourism. The reputation and recognition of the place as a gastronomic reference, as well as the roots and tradition of the wineries, are the elements that can improve the intention of wine tourism. This insight can guide management strategies in wineries and contribute to the tourism policy of Baja California, potentially increasing the number of wine tourists in the region.

## 7. Limitations and Future Research

One of the main limitations of the research was the sampling location since, as mentioned above, Baja California is the main area of wine production in Mexico, but it is not the only one. Future research can expand this research throughout all the wine regions of the country and thus be able to find similarities or differences between the regions, which could give recommendations for the regions or the entire country of Mexico. Thus, it is crucial to consider the findings of this research with caution since they only represent visitors in the Baja California region. Future studies can also conduct research among various wine-producing countries to explore the characteristics of wine tourists in each country.

Another important limitation was the homogeneity in the size of the groups, since the group of women predominated over the group of men. Although groups were not compared, and some authors such as Dewi [42] mention that each group should have at least 30 data records, it is necessary to take the results with caution since they are representative of segments with sociodemographic characteristics based on the sample. It is highly recommended that future studies consider sampling strategies to provide greater diversity of groups and homogeneity in the size of each one.

One limitation of the study was the selection of the type of sampling. Convenience sampling facilitates the collection of data from a group with particular characteristics, which was what the research sought. It was easier and faster to ask those willing to visit a winery in Baja, California directly through this type of sampling than by other methods, such as stratified or random sampling. However, this limits the generalization of the results, though the sample size ensures its representativeness to make inferences about populations with similar characteristics to those in the sample.

Given that this study is exploratory in nature and the model incorporates only three independent variables that are identified as significant according to the results, it offers merely an approximation of the factors influencing the intention to engage in wine tourism. It is imperative to consider additional elements, such as pricing, the attractiveness of the destination, quality, and overall experience. Including these factors could enhance the model’s explanatory power and more effectively capture the complexity of this phenomenon.

## Figures and Tables

**Figure 1 foods-13-03651-f001:**
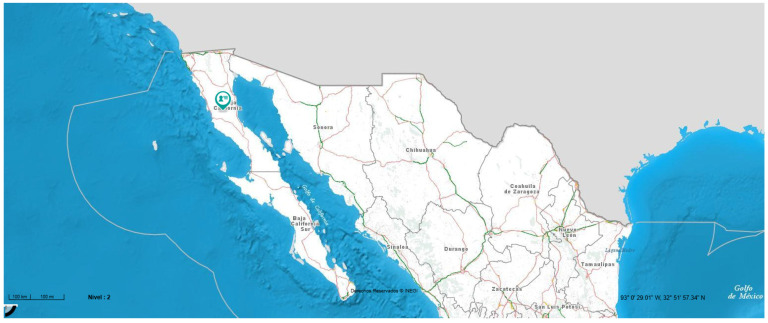
Map of Baja California. Source: [28].

**Figure 2 foods-13-03651-f002:**
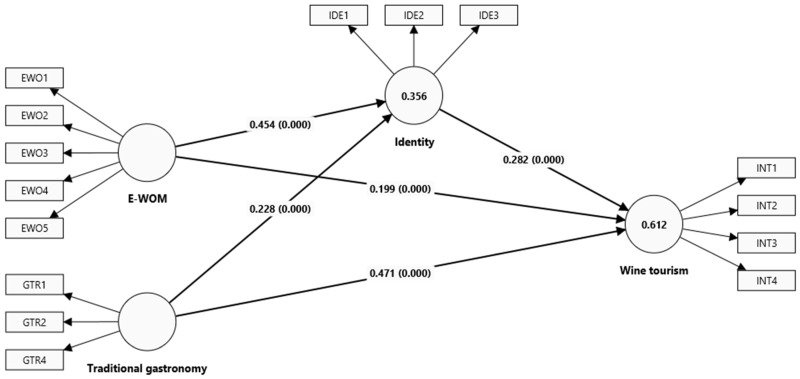
Structural model of wine tourism intention.

**Figure 3 foods-13-03651-f003:**
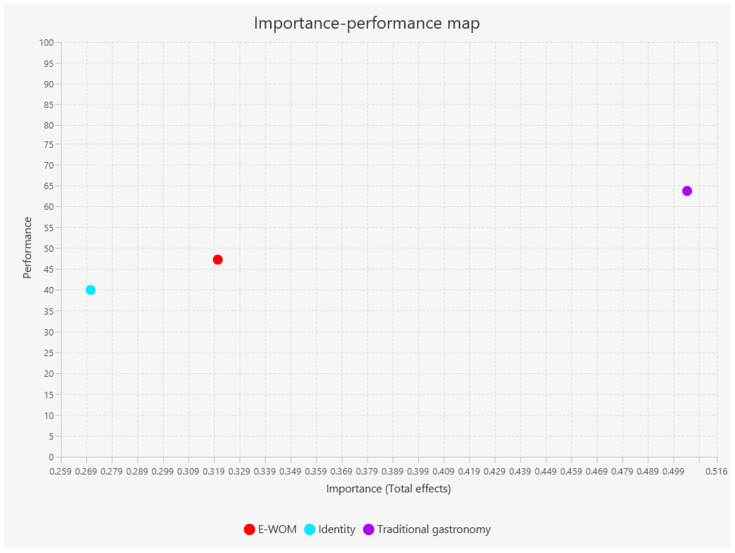
Importance–performance map at the variable level.

**Figure 4 foods-13-03651-f004:**
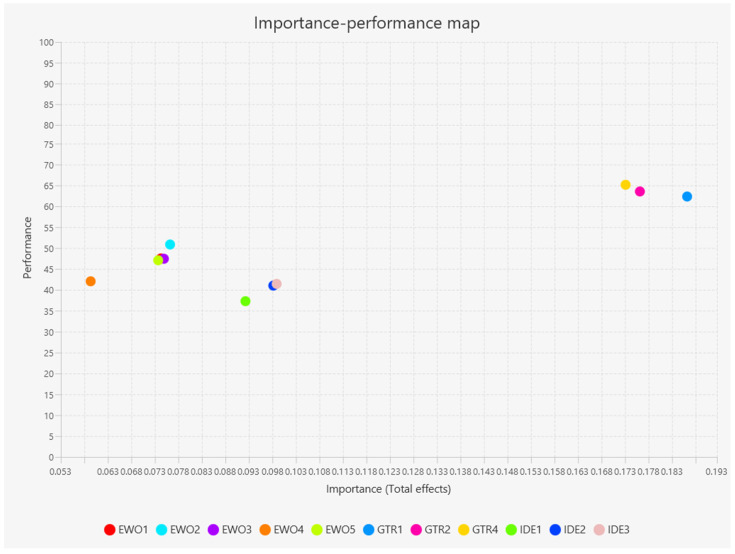
Importance–performance at the indicator level.

**Table 1 foods-13-03651-t001:** Variable measurements.

Variable		Author
E-WOM		[31]
EWO1	To make sure I choose the right winery, I read online reviews.	
EWO2	I check online reviews of food and wine places to help me choose my meal.	
EWO3	I gather information from wine reviews before I buy a wine.	
EWO4	I worry about my choice if I don’t read online wine reviews when I visit a winery.	
EWO5	When I go shopping, online wine reviews give me confidence in my decision.	
Traditional gastronomy		[32]
GTR1	Baja California is known for its wineries.	
GTR2	Baja California is known for its gastronomic heritage.	
GTR3 *	Baja California’s wineries have a good reputation.	
GTR4	Baja California’s wineries have a tradition and roots in the local population.	
Identity		[16]
IDE1	Being a connoisseur of Baja California wines is essential to who I am.	
IDE2	I regularly think about the characteristics of Baja California wines.	
IDE3	I need to be a connoisseur of Baja California wines.	
Wine tourism intention		[33]
INT1	I would look for Baja California wineries to visit.	
INT2	I recommend visiting Baja California wineries.	
INT3	I would be a loyal buyer of Baja California wineries.	
INT4	I would continue to visit Baja California wineries.	

* Item not validated.

**Table 2 foods-13-03651-t002:** External loads of items.

Item	External Loads
EWO1 <- E-WOM	0.887
EWO2 <- E-WOM	0.911
EWO3 <- E-WOM	0.895
EWO4 <- E-WOM	0.71
EWO5 <- E-WOM	0.881
GTR1 <- Traditional gastronomy	0.949
GTR2 <- Traditional gastronomy	0.898
GTR4 <- Traditional gastronomy	0.882
IDE1 <- Identity	0.859
IDE2 <- Identity	0.914
IDE3 <- Identity	0.921
INT1 <- Wine tourism intention	0.892
INT2 <- Wine tourism intention	0.911
INT3 <- Wine tourism intention	0.847
INT4 <- Wine tourism intention	0.907

**Table 3 foods-13-03651-t003:** Construct reliability and convergent validity.

Latent Variable	Cronbach’s Alpha Coefficient	Composite Reliability (rho_a)	Composite Reliability (rho_c)	Average Variance Extracted (AVE)
E-WOM	0.933	0.939	0.934	0.739
Identity	0.926	0.927	0.926	0.807
Traditional gastronomy	0.935	0.936	0.935	0.828
Wine tourism intention	0.938	0.939	0.938	0.792

**Table 4 foods-13-03651-t004:** Heterotrait–Monotrait Ratio.

Relationship	Ratio Heterotrait–Monotrait (HTMT)
Identity—E-WOM	0.565
Traditional gastronomy—E-WOM	0.482
Traditional gastronomy—Identity	0.448
Wine tourism intention—E-WOM	0.583
Wine tourism intention—Identity	0.606
Wine tourism intention—Traditional gastronomy	0.693

**Table 5 foods-13-03651-t005:** Structural model assessment.

Relationship	β	Mean	Standard Deviation	t	f^2^	VIF
E-WOM → Wine tourism intention	0.199 ***	0.199	0.04	4.917	0.063	1.627
Traditional gastronomy → Wine tourism intention	0.471 ***	0.472	0.035	13.557	0.415	1.387
E-WOM → Identity	0.454 ***	0.454	0.04	11.355	0.245	1.306
Traditional gastronomy → Identity	0.228 ***	0.228	0.039	5.843	0.062	1.306
Identity → Wine tourism intention	0.282 ***	0.281	0.039	7.266	0.132	1.557
E-WOM → Identity→ Wine tourism intention	0.128 ***	0.128	0.02	6.245	--	--
Traditional gastronomy → Identity → Wine tourism intention	0.064 ***	0.064	0.013	4.877	--	--

*** *p* < 0.001, β—path coefficient, t—t-value, f^2^—effect size, VIF—variance inflation factor, -- Not applicable.

## Data Availability

The original contributions presented in the study are included in the article, further inquiries can be directed to the corresponding author.

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
