# Peer review of "Motivators of the Intention of Wine Tourism in Baja California, Mexico"

_foods, 2024, doi:10.3390/foods13223651_

Round 1
Reviewer 1 Report
Comments and Suggestions for Authors
This manuscript, investigating wine tourism intention in Baja California, Mexico, utilizes PLS-SEM and IPMA but suffers from several methodological and interpretative shortcomings.
1.This study's reliance on a convenience sample of winery visitors in Baja California, predominantly female and Gen Z, severely limits the generalizability of its findings. These results cannot be confidently extrapolated to all Mexican wine tourists. The study acknowledges this limitation, but further justification for this sampling strategy is lacking, and more diverse geographical representations and age groups are needed to improve external validity.
2.While considering e-WOM and traditional gastronomy, the model neglects other potentially influential factors, such as price, service quality, destination attractiveness, and overall tourism experience. This omission compromises the model's explanatory power and fails to capture the complexity of the formation of wine tourism intention.
3. Although this study highlights traditional gastronomy as the most influential factor, the presentation of the maps lacks a detailed analysis and discussion of the findings, hindering the extraction of practical managerial implications. More specific strategic recommendations should be derived from the IPMA results.
4. While acknowledging the geographical constraints, this study overlooks the critical assumptions and limitations of PLS-SEM. Although PLS-SEM is less demanding regarding normality assumptions, its potential impacts should be discussed. Furthermore, mediation analysis requires more robust justification. Exploring alternative mediating variables and clearly explaining their mechanisms would enhance the clarity and validity of this claim. The study's conclusions also provide broad statements about policy implications, without sufficient supporting evidence.
Author Response
Comments 1: This study's reliance on a convenience sample of winery visitors in Baja California, predominantly female and Gen Z, severely limits the generalizability of its findings. These results cannot be confidently extrapolated to all Mexican wine tourists. The study acknowledges this limitation, but further justification for this sampling strategy is lacking, and more diverse geographical representations and age groups are needed to improve external validity.
Response 1: The representativeness of the groups within the methodology was justified and the limitation of the homogeneity of group size was highlighted.
Comments 2: While considering e-WOM and traditional gastronomy, the model neglects other potentially influential factors, such as price, service quality, destination attractiveness, and overall tourism experience. This omission compromises the model's explanatory power and fails to capture the complexity of the formation of wine tourism intention.
Response 2: This limitation is set out in Section 7.
Comments 3: Although this study highlights traditional gastronomy as the most influential factor, the presentation of the maps lacks a detailed analysis and discussion of the findings, hindering the extraction of practical managerial implications. More specific strategic recommendations should be derived from the IPMA results.
Response 3: Recommendations derived from the IPMA results were added in the discussion section.
Comments 4: While acknowledging the geographical constraints, this study overlooks the critical assumptions and limitations of PLS-SEM. Although PLS-SEM is less demanding regarding normality assumptions, its potential impacts should be discussed. Furthermore, mediation analysis requires more robust justification. Exploring alternative mediating variables and clearly explaining their mechanisms would enhance the clarity and validity of this claim. The study's conclusions also provide broad statements about policy implications, without sufficient supporting evidence.
Response 4: The mediation analysis process was detailed in the results section, and the data analysis by PLS-SEM was justified in the materials and methods section.
Reviewer 2 Report
Comments and Suggestions for Authors
The authors have put in some work, but this paper needs to be clarified in some parts.
Key words from the title of the paper are missing. They need to be reformulated.
Since the research was done in a certain geographical area, it is necessary to add a map, so that the readers know in which region the research was carried out.
The text states "to visit wine tourism". Wine tourism cannot be visited. We can engage in wine tourism or get involved as visitors.
This phrase is repeated in the text, so the authors should distinguish whether they are writing about participation as producers of wine tourism, or as tourists who visit wineries or travel along wine routes.
The authors should make a distinction between discussing involvement as wine tourism producers and visitors who visit vineyards or travel along wine routes because this phrase appears often in the text.
Introduction
Line 27/103
The writers tried to incorporate a literature review and some methodology in the introduction. The introduction should be modified.
A separate review of the literature chapter needs to be added.
Line 35/39
Describe the cause of it.
Line 52
Prior to the words "These wine routes" being used, there was no reference of any wine routes in the text.
Line 212
The discussion-related section of the text needs to be upgraded both qualitatively and scientifically. It is necessary to provide readers with a more comprehensive description of the outcomes that were obtained.
Conclusions
The conclusion needs to be expanded considering the study that was done. The authors should clarify the results of their research to complete the readers' understanding of the previous text.
Authors should write, maybe in a new chapter, an overview of the obstacles that existed during the research, as well as future considerations.
Author Response
Comments 1: Key words from the title of the paper are missing. They need to be reformulated.
Response 1: The keywords from the title were added.
Comments 2: Since the research was done in a certain geographical area, it is necessary to add a map, so that the readers know in which region the research was carried out.
Response 2: Map added
Comments 3: The text states "to visit wine tourism". Wine tourism cannot be visited. We can engage in wine tourism or get involved as visitors. This phrase is repeated in the text, so the authors should distinguish whether they are writing about participation as producers of wine tourism, or as tourists who visit wineries or travel along wine routes. The authors should make a distinction between discussing involvement as wine tourism producers and visitors who visit vineyards or travel along wine routes because this phrase appears often in the text.
Response 3: The paragraph was rewritten.
Comments 4:
Introduction
Line 27/103
The writers tried to incorporate a literature review and some methodology in the introduction. The introduction should be modified.
A separate review of the literature chapter needs to be added.
Line 35/39
Describe the cause of it.
Baja rentabilidad, escaza investigación y transferencia de tecnología, consolidación de una política agropecuaria, asesoría en la transformación, integración y fortalecimiento de actividades agropecuarias
Line 52
Prior to the words "These wine routes" being used, there was no reference of any wine routes in the text.
Response 4: A literature review section was included, and comments were addressed.
Comments 5:
Line 212
The discussion-related section of the text needs to be upgraded both qualitatively and scientifically. It is necessary to provide readers with a more comprehensive description of the outcomes that were obtained.
Response 5: The discussion was rewritten according to your recommendations.
Comments 6:
Conclusions
The conclusion needs to be expanded considering the study that was done. The authors should clarify the results of their research to complete the readers' understanding of the previous text.
Authors should write, maybe in a new chapter, an overview of the obstacles that existed during the research, as well as future considerations.
Response 6: A new Limitations and Future Research section was added. Additionally, the conclusion was rewritten to clarify the results.
Reviewer 3 Report
Comments and Suggestions for Authors
· Explain or include why convenience sampling was used, which may limit generalizability. Stratified or randomized sampling method could improve the representation of various demographic segments.
· Convergent and discriminant validity have been well-assessed; however, adding confirmatory factor analysis results will provide further reliability for the construct validity.
· Effect sizes are discussed, but it would be helpful to relate them to practical significance and provide a more detailed interpretation of moderate and small effect sizes in the context of wine tourism.
· While the R² and Q² values indicate model strength, additional interpretation of these values in terms of practical relevance would be valuable.
· The introduction provides useful background information, though some points are redundant (e.g., repeated mentions of the importance of traditional gastronomy). A more focused lead-in on specific research gaps would improve engagement.
· Streamlining the conclusion to highlight only novel implications will strengthen the impact of study.
· Emphasize on how this research advances understanding beyond existing studies, such as its contribution to tourism policy or consumer engagement in Baja California.
Overall, the manuscript is well-structured and covers a significant topic, addressing the areas of improvement outlined will enhance clarity, impact, and scholarly rigor. This is a promising study that would benefit from major revisions to strengthen interpretation and presentation.
Author Response
Comments 1: Explain or include why convenience sampling was used, which may limit generalizability. Stratified or randomized sampling method could improve the representation of various demographic segments.
Response 1: The sampling method was added as limitations.
Comments 2: Convergent and discriminant validity have been well-assessed; however, adding confirmatory factor analysis results will provide further reliability for the construct validity.
Response 2: SRMR values added
Comments 3: Effect sizes are discussed, but it would be helpful to relate them to practical significance and provide a more detailed interpretation of moderate and small effect sizes in the context of wine tourism.
Response 3: These effects were discussed in the manuscript.
Comments 4: While the R² and Q² values indicate model strength, additional interpretation of these values in terms of practical relevance would be valuable.
Response 4: The results were revisited during the discussions.
Comments 5: The introduction provides useful background information, though some points are redundant (e.g., repeated mentions of the importance of traditional gastronomy). A more focused lead-in on specific research gaps would improve engagement.
Response 5: The introduction was rewritten according to your comments and emphasized how the study addresses gaps in the literature.
Comments 6: Streamlining the conclusion to highlight only novel implications will strengthen the impact of study.
Response 6: The conclusion was rewritten according to your comments.
Comments 7: Emphasize on how this research advances understanding beyond existing studies, such as its contribution to tourism policy or consumer engagement in Baja California.
Response 7: The contribution of the study was highlighted in the conclusions
Round 2
Reviewer 1 Report
Comments and Suggestions for Authors
Dear authors,
I appreciate your detailed reply to my feedback, and for integrating the proposed suggestions, I have reviewed the changes closely and am pleased to see that my issues have been adequately addressed. I believe that the manuscript has been significantly improved and is now being prepared for publication.
Reviewer 3 Report
Comments and Suggestions for Authors
I do not have any further comments and appreciate authors for making significant changes in the manuscript.